# Analytical Modeling and Experimental Validation of an Energy Harvesting System for the Smart Plate with an Integrated Piezo-Harvester

**DOI:** 10.3390/s19040812

**Published:** 2019-02-16

**Authors:** Andrzej Koszewnik

**Affiliations:** Faculty of Mechanical Engineering, Bialystok University of Technology, Wiejska 45C, 15-351 Bialystok, Poland; a.koszewnik@pb.edu.pl

**Keywords:** smart SFSF plate, energy harvesting, Kirchhoff plate theory

## Abstract

The literature on piezoelectric energy harvesting (PEH) is strongly focused on structures, like cantilever beams with piezoceramic layers, due to the fact that they are easily modelled and implemented. As compared to the number of studies dealing with the aforementioned case, research on 2D structures with an attached piezoceramic patch harvester is very limited. Thus, an analytical modeling and experimental validations of a piezo harvester structurally integrated on a thin plate with SFSF (Simply supported-Free-Simply supported-Free) boundary conditions is presented in this paper. The distributed parameter electroelastic model of a harvester bonded to an aluminum plate with both piezo-patch actuators is developed on the basis of the Kirchhoff plate theory and the modal analysis for physical and modal coordinates. This allows to estimate the steady-state value output voltage for each odd mode in the frequency range of 10–300 Hz. Finally, the obtained results for the electroelastic analytical model is experimentally verified on a laboratory stand.

## 1. Introduction

Vibration-based energy harvesting systems have been extensively studied by many researches over the past two decades [1,2,3]. Many of them used electrostatic [4,5], piezoelectric [6,7], magnetostrictive [8] or electromagnetic [9,10] conversion techniques to transform available ambient energy into electrical energy. Among the aforementioned techniques, the piezoelectric one is the most commonly applied method in some applications to active vibration control or structural health monitoring (SHM) of civil structures [11]. 

The literature on Piezoelectric Energy Harvesting (PEH) is strongly focused on structures, like cantilever beams with piezoceramic layers, due to the fact that they are easily modelled and implemented [12,13,14,15]. Analytical and numerical models of structures with an attached piezo-harvester have been developed by several research groups. For instance, analytical distributed parameter modeling of beams for chaotic vibration with experimental set-up were presented by Litak [16], modeling of the beam based on Rayleigh–Ritz solutions excited to vibration by random signals was described by Erturk [17], while modeling of a self-resonating energy harvester system of cantilever beams with identification and experimental investigations was widely carried out by Wallasheck [18]. Another application of a piezo-harvester is shown in the paper published by Zhu who applied the integrated PZT sensor to the structure and next used them to monitor gas pipelines as a novel technology for leak detection [19]. 

As compared to the number of studies dealing with piezoelectric harvester beams, research on 2D structures with an attached piezoceramic patch harvester is very limited. For instance, Marqui presented an electromechanical finite element model for a PEH embedded in a cantilever plate, and later extended this model to airflow excitation problems by electroelastic coupling for energy harvesting from aeroelastic flutter [20]. A similar application can be found in the paper published by Anton, who was the first to design and investigate novel piezoelectric devices installed on UAV platforms. For instance, the author showed the piezoelectric patch with a thin-film battery as a multifunctional self-charging device for scavenging energy in reference [21]. Later, the same author presented a hybrid device containing piezo-electric stripes, macro-fiber and piezo-fiber composites that allows to harvest energy from wing vibrations [22]. The obtained results from these investigations lead to further development of this research area and their applications for civil structures. For instance, Harne modeled electroelastic dynamics of a vibrating panel with a corrugated piezoelectric spring and next analyzed this corrugated harvester device by attaching it to a panel of a public bus [23,24]. In other papers authors used vibration energy harvesting devices for monitoring a full-scale bridge as a 2D structure undergoing forced dynamic vibrations generated by a vehicle passage across bridges [25,26]. The additional advantage of this is the fact that the harvested energy can be used to sufficiently power other small devices with low power demand. 

The obtained results from the above papers and monitoring of civil infrastructure (bridge) by using longitudinally located harvester have been a motivation to analyze the harvester locations and orientations on the structure in order to obtain effective energy harvesting. For this purpose, the piezo-sensor used to measure vibration of the SFSF (Simply supported-Free-Simply supported-Free) plate in paper [27] was replaced by a piezo harvester. This leads to obtain a smart structure complex with two piezo-patch actuators and one harvester that are used to determine an electromechanical model based on Kirchhoff plate theory [28]. In order to determine piezo-harvester locations and orientations on the structure, the numerical analysis of mechanical strains was performed in Ansys software for the frequency range up to 300 Hz containing the first five natural frequencies of the structure. The indicated harvester locations and orientations allowed us to calculate a modal electromechanical coupling term and assess the value of the DC voltage from the harvester for two different forces generated by a piezo-actuator oriented in two perpendicular directions. An experimental test carried out in the lab properly verified the results calculated for the electroelastic analytical model, and indicated that the harvester orientation has its impact on DC voltage. The conclusions provided in the last section indicate that the piezo-harvester and piezo-actuator orientations have their impact on DC voltage and can be used to power small devices mounted on UAV or monitor civil infrastructures. 

Finally, it can be noticed that the proposed method is a modern approach to investigate energy harvesting systems, because unlike others, additional piezo-actuator orientation was also considered. This approach can be especially important in two fields: aerospace and building, where proper determination of harvester orientation and piezo-stripe actuator orientation generating maximum excitation force in a chosen frequency range can result in better efficiency of energy harvesting systems.

## 2. Electromechanical Model of the Plate: An Analytical Approach

In this section, a brief description of an electromechanical model distributed-parameters of a smart plate with piezo-patches is presented, based on the Kirchhoff plate theory. The aluminum SFSF plate, which represents the host structure, is equipped with two piezo-patch actuators and one piezo harvester. As a result, a smart structure, shown in Figure 1, is obtained. Parameters of the plate and piezoceramic patches adjusted to its surface are collected in Table 1.

The whole plate with integrated piezoceramic patches is excited to vibration by bending moment *Mx(x,y)* or *My(x,y)* at the position of both actuators. The perfectly bonded actuators have the same length (*l_as_*), width (*w_as_*) and thicknesses (*h_a_*). The first actuator PA_1 covers some regions of the plate surface at its corners (*x*_*a*1_,*y*_*a*1_) and (*x*_*a*2_,*y*_*a*2_), but the second PA_2 at—corners (*x*_*a*3_,*y*_*a*3_) and (*x*_*a*4_,*y*_*a*4_), respectively. The piezo-harvester (PEH) with the length of (*l_peh_*), the width of (*w_peh_*) and the thickness of (*h_peh_*) was additionally integrated with the structure at the positions (*x*_*h*1_,*y*_*h*1_) and (*x*_*h*2_, *y*_*h*2_) in order to measure voltage from vibrations. A resistive load *R*, shown in Figure 1, is considered as an external electrical load connected to the conductive electrode layers with their negligible thicknesses of the covered surfaces of the harvester. As a result, a piezo-stripe element is coupled with the host structure (only electromechanically). 

The case of this excited 2D mechanical structure leads to formulating the general Equationof transverse vibration of the smart plate in the following form [27]:(1)$$\begin{array}{l} \frac{\partial^{2}\left(M_{1}{}^{plate}+M_{1}{}^{peh}\right)}{\partial x^{2}}+2\frac{\partial^{2}\left(M_{6}{}^{plate}+M_{6}{}^{peh}\right)}{\partial x\partial y}+\frac{\partial^{2}\left(M_{2}{}^{plate}+M_{2}{}^{peh}\right)}{\partial y^{2}}+\\ -c\frac{\partial w(x,y,t)}{\partial t}-\rho_{plate}h_{plate}\frac{\partial^{2}w(x,y,t)}{\partial t^{2}}+\left[\frac{\partial^{2}M_{x}(x,y)}{\partial x^{2}}+\frac{\partial^{2}M_{y}(x,y)}{\partial y^{2}}\right]=0 \end{array}$$
where:*M*_1_^plate^, *M*_1_*^peh^*—internal bending moments of the plate and PEH in *X* direction, *M*_2_^plate^, *M*_2_*^peh^*—internal bending moments of the plate and PEH in *Y* direction, *M*_6_^plate^, *M*_6_*^peh^*—internal bending moments of the plate and PEH in in *X-Y* plane,*c*—the viscous damping coefficient,*M_x_*(*x,y*), *M_y_*(*x,y*)—the bending moment generated by the piezo-actuator PA_1 and PA_2 in *X* and *Y* direction, respectively, *w(x,y,t)*—vertical deflection of the plate at position (*x,y*) and time *t*.

The bending moments written in Equation (1) associated to the host plate and the piezo-harvester element can be expressed in the following forms: (2)$$M_{1}{}^{plate}=-D\left(\frac{\partial^{2}w\left(x,y,t\right)}{\partial x^{2}}+\nu \frac{\partial^{2}w\left(x,y,t\right)}{\partial y^{2}}\right)$$
(3)$$M_{2}{}^{plate}=-D\left(\frac{\partial^{2}w\left(x,y,t\right)}{\partial y^{2}}+\nu \frac{\partial^{2}w\left(x,y,t\right)}{\partial x^{2}}\right)$$
(4)$$M_{6}{}^{plate}=-D\left(1-\nu \right)\frac{\partial^{2}w\left(x,y,t\right)}{\partial x\partial y}$$
(5)$$M_{1}{}^{peh}=\left[H(x-x_{h1})-H(x-x_{h2})\right]\left[H(y-y_{h1})-H(y-y_{h2})\right]\times \int_{peh}T_{1}{}^{peh}zdz$$
(6)$$M_{2}{}^{peh}=\left[H(x-x_{h1})-H(x-x_{h2})\right]\left[H(y-y_{h1})-H(y-y_{h2})\right]\times \int_{peh}T_{2}{}^{peh}zdz$$
(7)$$M_{6}{}^{peh}=\left[H(x-x_{h1})-H(x-x_{h2})\right]\left[H(y-y_{h1})-H(y-y_{h2})\right]\times \int_{peh}T_{6}{}^{peh}zdz$$
where:
*D*—the flexural rigidity of the considered plate $D=\frac{E_{plate}h_{plate}}{12(1-v^{2})}$,*v*—the Poisson ratio of aluminum plate,*H(x)*, *H(y)*—the Heaviside functions,*T*_1_*^peh^*—normal stress of the harvester along *X* axis, *T*_2_*^peh^*—normal stress of the harvester along *Y* axis,*T*_6_*^peh^*—shear stress in the *X-Y* plane. 

The obtained moments of the host structure and the piezo-harvester in Equations (2)–(7) put into Equation (1) lead to determine the differential equation of the smart structure with an integral harvester. Then, Equation (1) is transformed to the following form: (8)$$\begin{array}{l} D\left(\frac{\partial^{4}w(x,y,t)}{\partial x^{4}}+2\frac{\partial^{4}w(x,y,t)}{\partial x^{2}y^{2}}+\frac{\partial^{4}w(x,y,t)}{\partial x^{4}}\right)+c\frac{\partial w(x,y,t)}{\partial t}+\rho_{harv}h_{peh}\frac{\partial^{2}w(x,y,t)}{\partial t^{2}}+\\ -\Gamma V_{p}(t)\left\{\begin{array}{l} \left[\frac{d\delta (x-x_{h1})}{dx}-\frac{d\delta (x-x_{h2})}{dx}\right]\times \left[H\left(y-y_{h1}\right)-H\left(y-y_{h2}\right)\right]+\\ \left[\frac{d\delta (y-y_{h1})}{dy}-\frac{d\delta (y-y_{h2})}{dy}\right]\times \left[H\left(x-x_{h1}\right)-H\left(x-x_{h2}\right)\right] \end{array}\right\}=\left[\frac{\partial^{2}M_{X}(x,y)}{\partial x^{2}}+\frac{\partial^{2}M_{Y}(x,y)}{\partial y^{2}}\right] \end{array}$$
where:
*ρ_p_*—mass density of the harvester,*h_peh_*—thickness of the harvester,*δ(x), δ(y)*—the Dirac delta function along the *X* and *Y* axes, respectively, *V_p_(t)*—voltage across the external resistive load *R*,Γ—the electromechanical coupling effect ($\Gamma =e_{31}(\frac{h_{plate}+h_{peh}}{2})$)

The indicated orientations of piezo-actuators integrated also with the surface of the 2D mechanical structure influence generating individual bending moments. Each mentioned moment is described by proper forces shown in Figure 1. As a result, the bending moment $M_{X}(x,y)$, acting longitudinally to *X* axis, has been calculated in reference to Equation (9a), while the moment $M_{Y}(x,y)$—according to Equation (9b).
(9a)$$M_{X}(x,y)=\frac{C_{0}\cdot V_{A}\cdot d_{31}}{h_{as}}\cdot \left[Q_{XX}\left(-\delta \left(x-x_{as1}\right)-\delta \left(x-x_{as2}\right)+2\delta \left(x-\frac{x_{as2}-x_{as1}}{2}\right)\right)\right]$$
(9b)$$M_{Y}(x,y)=\frac{C_{0}\cdot V_{A}\cdot d_{32}}{h_{as}}\cdot \left[Q_{YY}\left(-\delta \left(y-y_{as3}\right)-\delta \left(y-y_{as4}\right)+2\delta \left(y-\frac{y_{as4}-y_{as3}}{2}\right)\right)\right]$$
where:*δ*—Dirac’s function,*V_A_*—voltage applied to the piezo-actuator,*h_as_*—thickness of the piezo-actuator,*d*_31_, *d*_32_—piezo-electric constants [29],*C*_0_—electromechanical coupling coefficient of the actuator [29].

The piezo-harvester integrated with the host structure requires considering the issue also from the electrical point of view. For this purpose, the electric model of this element has been determined on the basis of the electric current which flows by the resistive load (*R*) applied to the system. As a result, the aforementioned current has the following form: (10)$$\frac{d}{dt}\int_{A}D_{e}\cdot ndA=\frac{V_{p}(t)}{R}$$
where:*D_e_*—electric displacement vector*n*—unit vector outward from the electrode surface.

As it was pointed out in the paper published by Erturk, the electric displacement of the piezo-harvester located on a 2D mechanical structure depends on their axial strain components $S_{1}^{P}$ and $S_{2}^{P}$ in the *X-Y* plane and the electrical field *E*_3_ in the vertical *Z*-axis [28]. This allows to express the electric displacement in the following form:(11)$$D_{e}={\bar{e}}_{31}S_{1}^{P}+{\bar{e}}_{32}S_{2}^{P}+{\bar{\varepsilon }}_{33}E_{3}$$
where:
$S_{1}^{P}(x,y,t)=-\left(\frac{h_{plate}+h_{peh}}{2}\right)\frac{\partial^{2}w(x,y,t)}{\partial x^{2}}$—axial strain in *X* axis, $S_{2}^{P}(x,y,t)=-\left(\frac{h_{plate}+h_{peh}}{2}\right)\frac{\partial^{2}w(x,y,t)}{\partial y^{2}}$—axial strain in *Y* axis, 

The obtained strains expressed in Equation (11) substituted to Equation (10) allow to determine an equation which governs the electrical circuit of the system in the form:(12)$$C_{p}\frac{dV_{p}(t)}{dt}+\frac{V_{p}}{R}+\Gamma \left[\int_{y_{h1}}^{y_{h2}}\int_{x_{h1}}^{x_{h2}}\left(\frac{\partial^{3}w(x,y,t)}{\partial x^{2}\partial t}+\frac{\partial^{3}w(x,y,t)}{\partial y^{2}\partial t}\right)dxdy\right]=0$$
where:*C_p_*—capacitance of the piezo-patch $\left(C_{p}=\frac{{\bar{\varepsilon }}_{33}w_{peh}l_{peh}}{h_{peh}}\right)$.

Both Equation (1) and Equation (12) refer to the distributed-parameters electroelastic model of the piezo-patch harvester in physical coordinates. From the analytical point of view, this model is obviously correct, but taking in consideration the strategy control, it should be analyzed in modal coordinates. As a result, the vertical displacement *w*(*x,y,t*) of the host structure is expressed in the following form:(13)$$w(x,y,t)=\sum_{n=1}^{\infty }\sum_{m=1}^{\infty }\varphi_{mn}(x,y)\eta_{mn}(t)$$
where:
$\varphi_{mn}(x,y)$—mass-normalized eigenfunction,$\eta_{mn}(t)$—modal time response for the *mn*th mode shape.

The considered plate, according to Figure 1, as it has already been mentioned, has two simply supported edges oriented longitudinally to the *Y* axis. As a result eigenvectors of this structure, after consideration of the boundary conditions written in Equation (14) and split of geometric and time variables, can be expressed as:(14)$$w(x,y)=0;\;\frac{\partial^{2}w(x,y)}{\partial x^{2}}+\nu \frac{\partial^{2}w(x,y)}{\partial y^{2}}=0;\;\mathrm{for}\;x=0,\;L$$
(15)$$w(x,y)=\Theta_{n}\left[\left(A_{n}ch\alpha_{n}y+B_{n}sh\alpha_{n}y+C_{n}\alpha_{n}ch\alpha_{n}y+D_{n}\alpha_{n}sh\alpha_{n}y\right)\sin\alpha_{n}x\right]$$
where:
$\alpha_{i}=\frac{n\pi }{L}$, *n* is *n*-th mode shape.*A_n_, B_n_, C_n_, D_n_*—coefficients of the vertical deflection individually determined for each mode shape,Θ_*n*_—modal amplitude constant.

Substituting the obtained Equation (15) to Equation (1) leads to solving the eigenvalue problem of the smart plate for short circuit conditions (*R*→0). Then, the natural frequency *ω_mn_* of the structure is simplified to the following form:(16)$$\omega_{mn}=\frac{\lambda_{mn}\pi^{2}}{L^{2}}\sqrt{\frac{D}{\rho_{p}h_{p}}}$$
where:$\lambda_{mn}$—frequency parameter of an undamped plate.

Taking into account the modal analysis procedure of the 2D structure with adjusted to its surface piezo-elements, an electromechanical coupled ordinary differential equation for the modal time response *η_mn_* can be expressed in the following form [30]:(17)$$\frac{d^{2}\eta_{mn}(t)}{dt^{2}}+2\xi_{mn}\omega_{mn}\frac{d\eta_{mn}(t)}{dt}+\omega^{2}{}_{mn}\eta_{mn}(t)-{\tilde{\Gamma }}_{mn}v(t)=f_{mn}{\left(t\right)}_{1}+f_{mn}{\left(t\right)}_{2}$$
where:$\xi_{mn}$—modal damping ratio determined in the identification procedure.$f_{mn}{\left(t\right)}_{1}$—modal force derived from the piezo-actuator oriented parallel to *X* axis, $f_{mn}{\left(t\right)}_{2}$—modal force derived from the piezo-actuator oriented parallel to *Y* axis, 

Then, each modal force $f_{mn}{\left(t\right)}_{1}$ and $f_{mn}{\left(t\right)}_{2}$, specified in Equation (17), can be written in the following form:(18)$$\begin{array}{l} f_{mn}{\left(t\right)}_{1}=f_{m}{\left(t\right)}_{1}=\int_{0}^{W}\int_{0}^{L}f(t)\delta \left(x-x_{as1}\right)\delta \left(y-\frac{y_{as2}-y_{as1}}{2}\right)\varphi_{mn}(x,y)dxdy+\\ \int_{0}^{W}\int_{0}^{L}f(t)\delta \left(x-x_{as2}\right)\delta \left(y-\frac{y_{as2}-y_{as1}}{2}\right)\varphi_{mn}(x,y)dxdy+-2\int_{0}^{W}\int_{0}^{L}f(t)\delta \left(x-\frac{x_{as2}-x_{as1}}{2}\right)\delta \left(y-\frac{y_{as2}-y_{as1}}{2}\right)\varphi_{mn}(x,y)dxdy \end{array}$$
(19)$$\begin{array}{l} f_{mn}{\left(t\right)}_{2}=f_{n}{\left(t\right)}_{2}=\int_{0}^{W}\int_{0}^{L}f(t)\delta \left(x-\frac{x_{as4}-x_{as3}}{2}\right)\delta \left(y-y_{as3}\right)\varphi_{mn}(x,y)dxdy+\\ \int_{0}^{W}\int_{0}^{L}f(t)\delta \left(x-\frac{x_{as4}-x_{as3}}{2}\right)\delta \left(y-y_{as4}\right)\varphi_{mn}(x,y)dxdy+-2\int_{0}^{W}\int_{0}^{L}f(t)\delta \left(x-\frac{x_{as4}-x_{as4}}{2}\right)\delta \left(y-\frac{y_{as4}-y_{as3}}{2}\right)\varphi_{mn}(x,y)dxdy \end{array}$$
while the modal electromechanical coupling term ${\tilde{\Gamma }}_{mn}$ may be expressed as:(20)$${\tilde{\Gamma }}_{mn}=\Gamma \left[\int_{y_{h1}}^{y_{h2}}{\frac{\partial \varphi_{mn}(x,y)}{\partial x}|}_{x_{h1}}^{x_{h2}}dy+\int_{x_{h1}}^{x_{h2}}{\frac{\partial \varphi_{mn}(x,y)}{\partial y}|}_{y_{h1}}^{y_{h2}}dx\right]$$

The vertical deflection of the host plate with the bonded piezo-harvester described in Equation (13), put into Equation (12), leads to defining its electrical circuit equation in the following form:(21)$$C_{p}\frac{dV_{p}(t)}{dt}+\frac{V_{p}(t)}{R}-\sum_{m=1}^{\infty }\sum_{n=1}^{\infty }{\tilde{\Gamma }}_{mn}\frac{d\eta_{mn}(t)}{dt}=0$$

The performed analytical considerations of the plate with piezo-elements for physical coordinates (see Equation (8) and Equation (12)) and modal coordinates (see Equation (17) and Equation (21)) lead us to calculate the value of the rectified voltage response accumulated on the resistor in the steady-state. For this purpose, assuming the harmonic form of the force generated by the piezo-actuator, formed as $f(t)=F_{0}\sin\left(\omega t\right)=F_{0}e^{j\omega t}$, can also express the modal response *η_mn_* and voltage response *V*(*t*) in the following form: (22)$$\eta_{mn}=H_{mn}\sin\left(\omega t\right)=H_{mn}e^{j\omega t};\;V(t)=V_{p}\sin\left(\omega t\right)=V_{p}e^{j\omega t}$$
where:
*F*_0_—amplitude of the force generated by the piezo-actuator,*ω*—excitation frequency,

Putting Equation (22) into Equation (17) and Equation (21) leads to obtaining the following equations of the electromechanical model and the electric circuit system:
(23)$$\begin{array}{l} -\omega^{2}H_{mn}e^{j\omega t}+2j\xi_{mn}\omega_{mn}\omega H_{mn}e^{j\omega t}+\omega_{mn}{}^{2}H_{mn}e^{j\omega t}-{\tilde{\Gamma }}_{mn}V_{p}e^{j\omega t}=F_{0}e^{j\omega t}\varphi_{mn}\left(x_{as1,}\frac{y_{as2}-y_{as1}}{2}\right)+\\ +F_{0}e^{j\omega t}\varphi_{mn}\left(x_{as2,}\frac{y_{as2}-y_{as1}}{2}\right)-2F_{0}e^{j\omega t}\varphi_{mn}\left(\frac{x_{as2}-x_{as1}}{2},\frac{y_{as2}-y_{as1}}{2}\right)+F_{0}e^{j\omega t}\varphi_{mn}\left({\frac{x_{as4}-x_{as3}}{2}}_{,}y_{as3}\right)+\\ +F_{0}e^{j\omega t}\varphi_{mn}\left({\frac{x_{as4}-x_{as3}}{2}}_{,}y_{as4}\right)-2F_{0}e^{j\omega t}\varphi_{mn}\left(\frac{x_{as4}-x_{as3}}{2},\frac{y_{as4}-y_{as3}}{2}\right) \end{array}$$
(24)$$C_{p}V_{p}j\omega e^{j\omega t}+\frac{V_{p}e^{j\omega t}}{R}+e^{j\omega t}\sum_{m=1}^{\infty }\sum_{n=1}^{\infty }j{\tilde{\Gamma }}_{mn}\omega H_{mn}=0$$

Next, the modal amplitude of the piezo-harvester response is determined by excluding the harmonic part *e^jwt^* from Equation (23): (25)$$H_{mn}=\frac{A+{\tilde{\Gamma }}_{mn}V}{\omega_{mn}{}^{2}+2j\xi_{mn}\omega_{mn}\omega -\omega^{2}}$$
where:$\begin{array}{l} A=F_{0}\varphi_{mn}\left(x_{as1,}\frac{y_{as2}-y_{as1}}{2}\right)+F_{0}\varphi_{mn}\left(x_{as2,}\frac{y_{as2}-y_{as1}}{2}\right)-2F_{0}\varphi_{mn}\left(\frac{x_{as2}-x_{as1}}{2},\frac{y_{as2}-y_{as1}}{2}\right)+F_{0}\varphi_{n}\left({\frac{x_{as4}-x_{as3}}{2}}_{,}y_{as3}\right)+\\ F_{0}\varphi_{mn}\left({\frac{x_{as4}-x_{as3}}{2}}_{,}y_{as4}\right)-2F_{0}\varphi_{mn}\left(\frac{x_{as4}-x_{as3}}{2},\frac{y_{as4}-y_{as3}}{2}\right) \end{array}$

As a result, the obtained Equation (25) put into Equation (24) results in expressing the modal voltage amplitude *V_p_* of the plate for each mode shape in the following form: (26)$$V_{p}\left(\omega \right)=\frac{-j\omega \sum_{m=1}^{N}\sum_{n=1}^{N}\frac{A{\tilde{\Gamma }}_{mn}}{\omega_{mn}{}^{2}+2j\xi_{mn}\omega_{mn}\omega -\omega^{2}}}{j\omega C_{p}+\frac{1}{R}+\sum_{m=1}^{N}\sum_{n=1}^{N}\frac{j\omega {\tilde{\Gamma }}_{n}{}^{2}}{\omega_{mn}{}^{2}+2j\xi_{mn}\omega_{mn}\omega -\omega^{2}}}=\frac{-j\omega \sum_{m=1}^{N}\sum_{n=1}^{N}\frac{A{\tilde{\Gamma }}_{mn}}{\omega_{mn}{}^{2}+2j\xi_{mn}\omega_{mn}\omega -\omega^{2}}}{j\omega C_{p}+\frac{1}{R}+\sum_{m=1}^{N}\sum_{n=1}^{N}\frac{j\omega {\tilde{\Gamma }}_{mn}{}^{2}}{\omega_{mn}{}^{2}+2j\xi_{mn}\omega_{mn}\omega -\omega^{2}}}$$

## 3. Numerical Analysis of Smart Plate

The numerical simulations of an aluminum plate with an integrated piezo-harvester and piezo-actuators were carried out with the use of Ansys and Matlab software. The main goal of these investigations was to obtain eigenvectors, eigenvalues and modal excitation forces generated by the piezo-actuators. For this purpose, taking into account analytical considerations from the previous section, the smart plate was modeled by using an FEM package. As a result, the host structure and piezo-elements are modeled as plane models PLANE 42. In contrast, the epoxy glue located between the host element and piezo-patches as a spring-damper is indicated as the COMBIN14 model of the following parameters: spring constant *K* = 100 kN and coefficient damping CV_1_ = 1000. 

Next, the numerical model of the considered structure was divided into 435 elements with the help of the meshing procedure. This leads to obtaining a singular finite element of the size of 25 mm and 12.5 mm, which corresponds with the half-size of the piezo-actuator. As the result of such discretization, the eigenvalue problem of the plate is solved and their results are shown in Figure 2. 

Taking into account Figure 2, it can be seen that the fundamental analytical natural frequency of the smart plate is 35.2 Hz and the corresponding mode shape is in-phase on the overall surface with the maximum deflection placed at the half-length of this structure (see Figure 2a). Higher mode shapes have several in-phase and out-of-phase regions across the surface of the plate. For instance, the second mode (*n* = 2, *f*_2_ = 69.5 Hz) has one node line at the center of the *Y* axis, the third mode (*f*_3_ = 137.2 Hz) has also one node line at the center, whereas in the *X* direction, the fifth mode (*f*_5_ = 220.6 Hz) has two node lines located longitudinally to the *Y* axis. The fourth mode (*f*_4_ = 164.5 Hz), where the nodal lines are located at the center of the plate in both directions *X* and *Y*, respectively, is a completely different case. 

The performed analysis of mode shapes in regard to the amount and direction of the nodes lines leads to assigning two kinds of modes, called “odd modes” and “even modes”, respectively. As a result, the group of “odd modes” is represented by only symmetrical modes located along the *Y* axis, while the “even modes”—by modes symmetrically only along the *X* axis or screw-symmetrically versus *X* and *Y* axis.

The carried out modal analysis allows to consider also a proper location of the harvester on the thin plate. For this purpose, the numerical investigations are repeated once again in order to determine the strain fields in two perpendicular directions of the plate, *X* and *Y,* respectively.

The obtained results (see Figure 3 and Figure 4) show that the strain mode shapes have node line regions in both directions apart from the fundamental mode of the considered plate. Considering the issue of energy harvesting, it is a significant problem because fast changes of the strain sign on the harvester electrode cause a strong reduction of the electrical output. As a result, the longitudinal harvester orientations on the plate to the Y axis are less beneficial, especially in the case of the multi-vibration analysis. The harvester location in (*X* = 0.10 m, *Y* = 0.05 m or *X* = 0.10 m, *Y* = 0.15 m) can be an exception, however, for considering only the second and the fourth modes, because the strain fields in these areas are in-phase or out-of-phase. Orienting the harvester longitudinally to the *X* axis would be a better solution in this case because the harvester can generate electrical outputs for almost all locations in the structure.

Finally, the piezo harvester with a negligible effect on the strain distribution is placed at the left-lower quadrant of the plate in the distance of *X* = 20 mm from the simply supported edge and *Y* = 50 mm from the free edge. 

The indicated harvester locations and orientations on the plate allow to calculate the electromechanical coupling factor Γ. For this purpose, with the help of Equation (20), the value of this factor was estimated for the determined mode shapes. The results are collected in Table 2. 

Determining the modal excitation forces was the next step in order to determine the output voltage from the piezo-harvester *V_p_*. For this purpose, the numerical model of the host structure with a piezo-actuator in this case has been used again. As it was described in the paper [26], proper orientation of actuators on the structure influences the value of the force generated by these elements. As a result, the possibilities of the energy harvesting system for the same actuators orientations are investigated. 

For this purpose, the calculations of the modal excitation force for chosen locations of piezo-actuators on the surface of the plate (see red rectangular in Figure 5 were carried out using Matlab and Ansys. Each time, it was assumed that the voltage applied to piezo-actuators electrodes had constant values that equaled 180 V. As a result, significant unit forces appeared in the indicated placement of the piezo-actuator. 

This allows to calculate modal excitation forces *f*_*n*_(*t*)_1_, *f*_*n*_(*t*)_2_ for piezo PA_1, according to Equation (18), and piezo PA_2, according to Equation (19), respectively.

The obtained results, shown in Figure 5 and collected in Table 3, indicate that orientations of a piezo-actuator on the 2D structure have a significant influence on the value of the forces generated by these elements. Taking into account the assumed boundary conditions of the considered structure (SFSF), it can be noticed that the most significant values of these forces are obtained for PA_1, and the lowest ones for PA_2. This resulted in the fact that further analysis regarding the harvesting energy system is performed for only odd modes. 

The damping vibrations coefficient was the last parameter needed to calculate the modal voltage. Its value was estimated on the basis of investigations performed for chosen piezo-actuator locations and described in paper [31]. Taking into account the identification procedure of the host plate with integrated piezo-elements, the mathematical model represented only by the odd modes is determined in the frequency range of 10–300 Hz. The obtained in Equation (27) reduced order model, that describes the transfer between the displacement and the force of the plate, is compared with an experimental plot recorded by using an analyzer (see Figure 6). In the results of the comparison of both magnitude plots (reduced order model and FRF), a good compatibility between them is achieved and the values of particular modal damping are obtained: ζ_1_ = 0.01, ζ_3_ = 0.005, ζ_5_ = 0.0007.
(27)$$\begin{array}{l} H_{odd\_\mathrm{mod}es}(s)=\frac{W_{odd\_\mathrm{mod}es}(s)}{F(s)}=\\ 0.0168\cdot \frac{\left(s^{2}+11.23s+6.482e4\right)\left(s^{2}+46.47s+9.353e5\right)\left(s^{2}+62.66s+2.685e6\right)}{\left(s^{2}+7.641s+4.885e4\right)\left(s^{2}+36.7s+8.443e5\right)\left(s^{2}+50.08s+2.574e6\right)} \end{array}$$

## 4. Standard AC-DC Problem of a Harvester with Experimental Validation

In this section, a standard AC-DC problem of an energy harvesting system for the mechanical model and also for a real structure is solved by connecting a full-wave rectifier with a smoothing capacitor (*C_p_*) and a resistive load (*R*) to the harvester. Figure 7 illustrates this connection. The AC-DC parameters model used in the simulation were assumed according to the datasheet of the EHE004 conditioning system that was used in the experimental set-up. As a result, the resistive load *R* equals 100 kΩ, while the smoothing capacitor *C_p_*—100 μF. 

The properties of the vibration-based energy harvesting system used in this case are estimated for the multi-mode analysis. The DC voltage is chosen as an indicator, which can be obtained from the mechanical model. The results of these investigations are shown in Figure 8.

The experiments carried out using a laboratory stand, shown in Figure 9, enabled us to verify the obtained results for the simulations. For this purpose, the SFSF smart plate with both piezo-actuators QP20N located in the indicated locations and a single piezo-harvester V21BL located close to the supported edge shown in Figure 9b is used as an example of a 2D structure. Apart from the plate, the laboratory stand has been equipped with a bipolar voltage amplifier SVR-150bip/3, a Digital Signal Analyzer (DSA) and the conditioning system EHE004, in order to test the energy harvesting system.

The starting point of the experimental set-up was exciting the structure to vibration by using piezo-actuators. For this purpose, the voltage signal has the form *u(t)* = 5sin(*ω_n_*t), where the *ω_n_* is chosen, the natural frequency of the structure (*f*_1_ = 35.2 Hz, *f*_3_ = 137.2 Hz, *f*_5_ = 252.0 Hz) is generated from DSA. Next, the obtained signal is amplified by the bipolar voltage amplifier and applied to the piezo PA_1. As a result, the force *F*_0_, as the excitation force, is generated from this piezo-patch element. From the harvesting point of view, vibrations of the structure are measured by the piezo-harvester. The obtained AC voltage signal is rectified by the conditioning system EHE004 and recorded with the help of the DSA. 

The experimental plots of DC voltage presented in Figure 10 (see black dot line, without shift (without consider offset value of DC voltage)) properly verified the numerical results in the time domain especially after crossing 100 s. This is especially visible only for the third and the fifth natural frequencies (8.5 mV for 3th mode and 0.45 V for 5th mode) where convergence between the measured response and the calculated voltage from electromechanical model is the highest. On the other hand, in the case of excitation plate with the first natural frequency, the recorded DC voltage from the test was less than 0.3 mV, therefore it was neglected by the analyzer. 

The obtained experimental plots also showed that the process of discharge of the filter capacitor *C_p_* was too short. As a result, it led to obtain two different offset values of DC voltages 6.2 mV for 3rd mode and 0.2 V for 5th mode, respectively. Taking into account this behavior, the voltage response of electromechanical model have been again calculated. For this purpose, two DC voltages of the model have been shifted up to offset value, while amplitudes of AC voltage given by Equation (26) have been calculated for lower values of damping coefficients. As a result, both responses of EH system and the electromechanical model have been adjusted to each other as it is shown in Figure 10 (see shifted plots). 

## 5. Discussion and Conclusions

The two-dimensional structures made of thin membranes or plates with various boundary conditions and integrated piezo-patches are commonly used in marine, aerospace and automotive applications. The aforementioned range of applications results in a variety of investigations concerning this kind of mechanical structure. Thus, in this work, the electroelastic parameter of a piezoelectric harvester located on the top surface of the host plate is developed and presented. Taking into account the chosen boundary conditions of a thin plate (SFSF), the possibilities of the considered mechanical structure and the equivalent model to harvest energy are investigated. For this purpose, as the first step, the analysis of mode shapes is carried out. The obtained results shown in Figure 2 allow to divide the considered modes into odd and even modes according to the appearing node lines. As a result of this conclusion, actuators which generate modal forces are located in their quasi-optimal locations from the low energy strategy point of view. The values of these forces obtained with the help of the FEM method indicated that the orientations of the piezo-actuators on the structure have a direct influence on the value of the excitation force generated by this element, but an indirect influence on the location of the harvester on the structure. As a result, the actuators have locations which generate maximum modal forces by minimum energy applied to the system. On the other hand, the piezo harvester is located in the vicinity of the simply supported edge of the structure, where the strains of the structure are significant. The determined harvester location on the thin plate and calculated modal values of electromechanical coupling factors (see Table 2) enabled us to obtain voltage *V_p_* for a mechanical model determined on the basis of the analytical approach. Estimating modal damping for the first three odd natural frequencies was an important stage of conducting this step of the research. This problem was solved by determining damping coefficients for the considered natural frequencies based on a mathematical model obtained from the identification procedure. 

Experimental investigations of the EH system, carried out on the laboratory stand for a real structure, also properly verified earlier results (see Figure 10). The recorded DC voltage signals from EHE004 system showed that its highest value is achieved for the plate excited to vibrations with the fifth natural frequency, but the lowest value - for the first natural frequency. This phenomenon is caused by the fact that bigger dynamic strains distributions appear along the shorter edge of the plate. Additionally, the recorded long times of increasing DC voltage result from the charging process of a smoothing capacitor *C_p_* used in the EHE004 system. 

Summarizing, the performed investigations of a piezoelectric patch harvester attached to thin plates enabled exploring the harvesting performance system located on 2D structures. Taking into account this fact, it can be noticed that the proposed method should be of a considerable interest to energy harvesting system engineers, who have to provide practical solutions, especially in the field of energy harvesting systems located on wings of UAVs, satellites or bridges. Then, properly determined piezo harvester and piezo-orientation on the structure can result in high efficiency of energy harvesting systems allowing powering of small devices mounted on UAVs or monitoring of civil infrastructures. 

Further investigations of these systems connected with non-linear electrical components of the equivalent circuit model can lead to an increase of applying these systems in various types of civil infrastructures, especially in a range of structural health monitoring or powering other small devices with low power demand. 

## Figures and Tables

**Figure 1 sensors-19-00812-f001:**
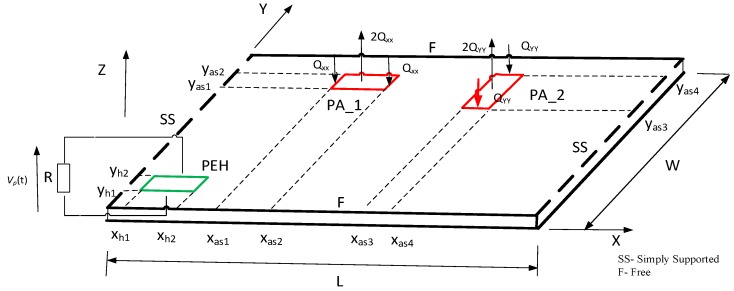
The rectangular aluminum plate with a piezo-harvester PEH and piezo-actuators PA_1 and PA_2 integrated on the top surface of this structure.

**Figure 2 sensors-19-00812-f002:**
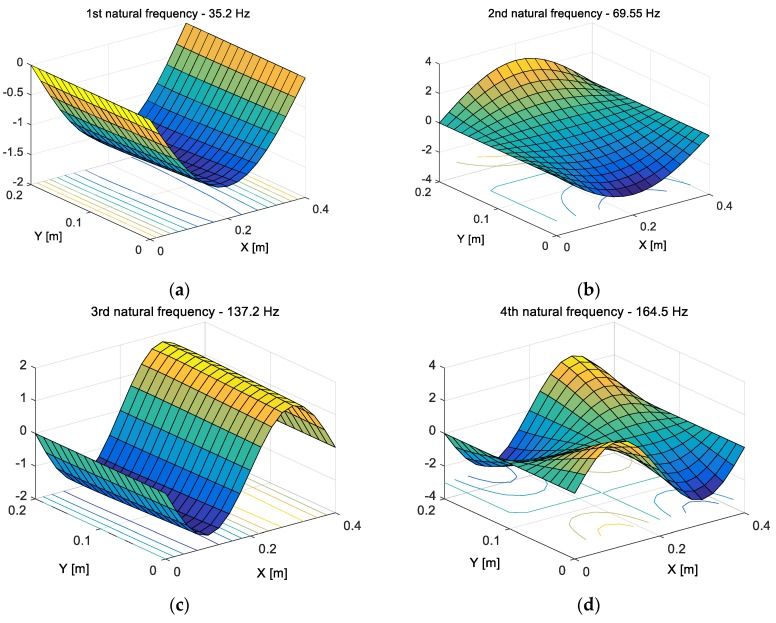
The first five mode shapes of the host structure with a piezo-harvester and piezo-actuators.

**Figure 3 sensors-19-00812-f003:**
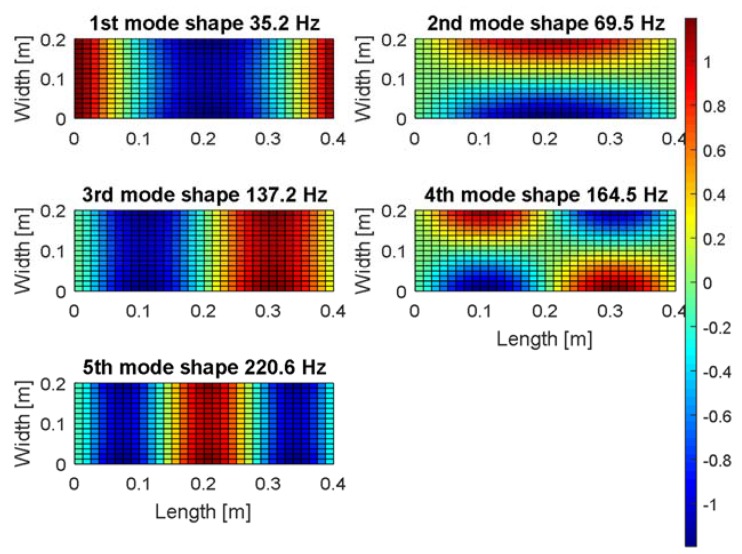
The mechanical strains fields in *X* direction of the smart SFSF plate with integrated piezo-elements for the first five mode shapes.

**Figure 4 sensors-19-00812-f004:**
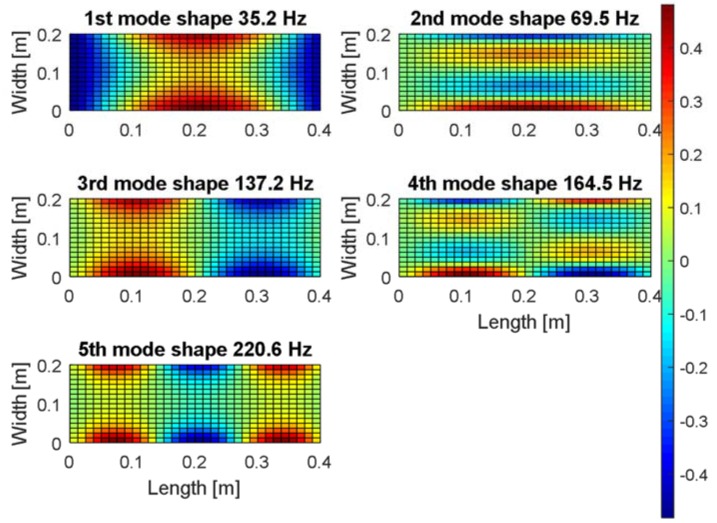
The mechanical strains fields in *Y* direction of the smart SFSF plate with integrated piezo-elements for the first five mode shapes.

**Figure 5 sensors-19-00812-f005:**
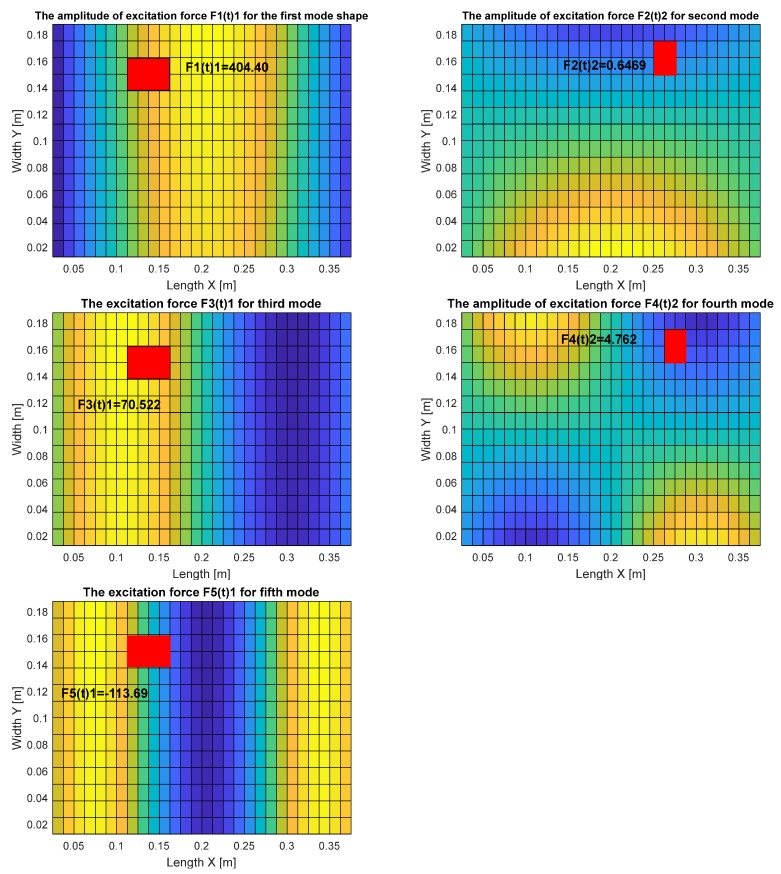
Maximum amplitude of the modal force calculated for piezo-actuator PA_1 and piezo-actuator PA_2 in the frequency range of 10–300 Hz.

**Figure 6 sensors-19-00812-f006:**
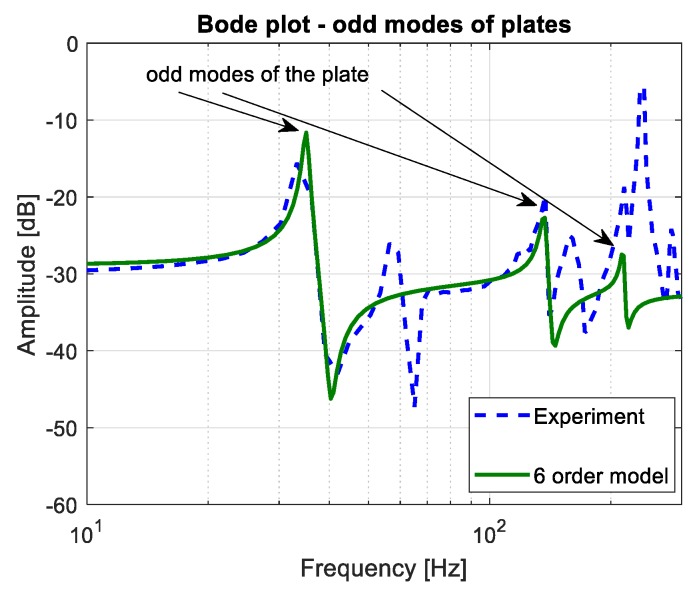
Comparison of the FRF of experimental data and 6th order model of the SFSF smart plate obtained for excitation by the piezo-stripe actuator oriented longitudinally to *X* axis (PA_1).

**Figure 7 sensors-19-00812-f007:**
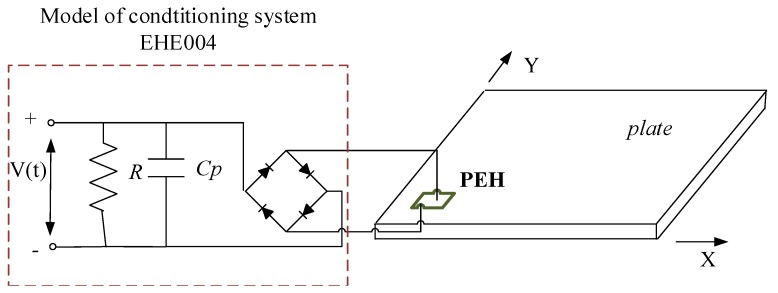
The piezo-harvester connected to a standard AC-DC conversion circuit: full-wave rectifier, a smoothing capacitor, resistive load.

**Figure 8 sensors-19-00812-f008:**
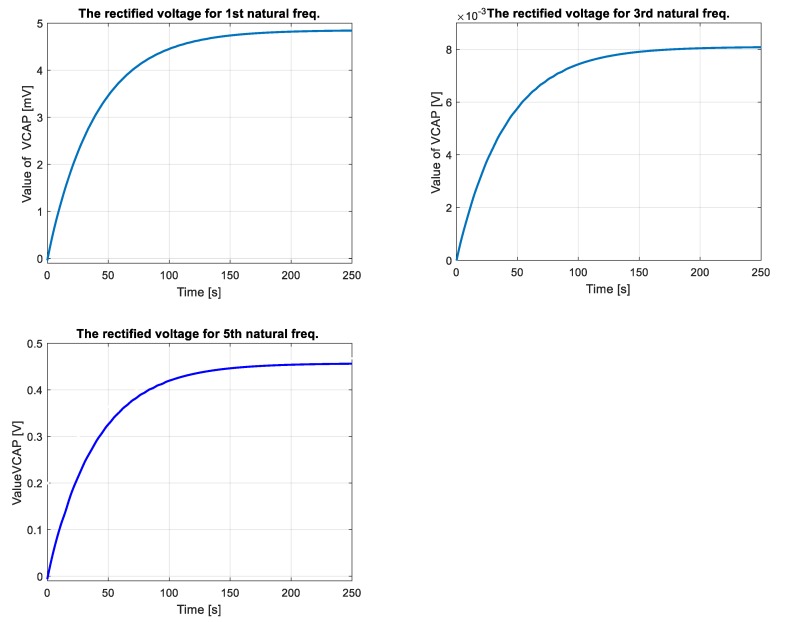
The rectified voltage from a full bridge rectifier with a smoothing capacitor *C_p_* = 400 μF and resistive load *R* = 100 kΩ obtained for the SFSF smart plate excited to vibration with the first three odd lowest natural frequencies.

**Figure 9 sensors-19-00812-f009:**
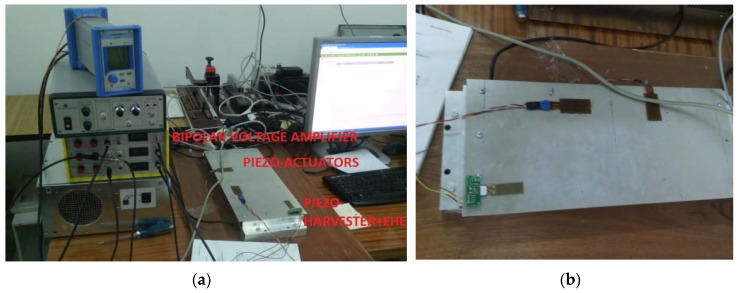
(**a**) The laboratory stand, (**b**) the smart plate with piezo-elements (actuators QP10N and harvester V21BL) located on its top surface.

**Figure 10 sensors-19-00812-f010:**
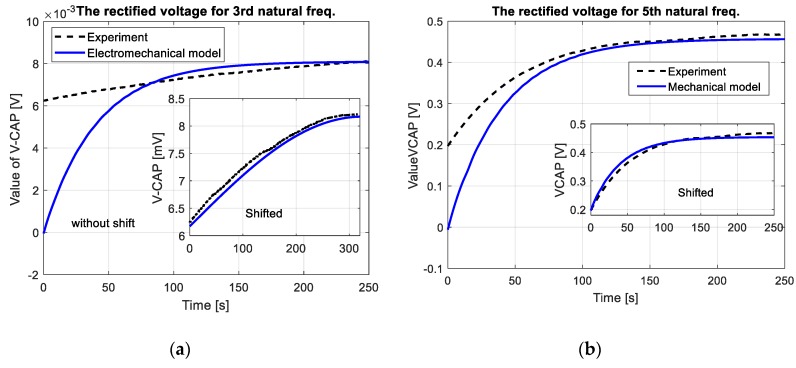
Comparison of numerical and experimental value of DC voltage obtained for the system excited to vibration with (**a**) 3rd natural frequency, (**b**) 5th natural frequency.

**Table 1 sensors-19-00812-t001:** Parameters of the host structure and piezo-elements.

Parameter	Plate		Piezo-Element	Actuator QP20N	Harvester V21BL
Length [m]	L	0.4	*l_p_/l_peh_*	0.05	0.048
Width [m]	*W*	0.2	*w_p_/w_peh_*	0.025	0.0125
Thickness [m]	*h_plate_*	0.002	*h_a_/h_peh_*	0.000782	0.000787
Young module [GPa]	*E_plate_*	70	*E_p_*	0.18	0.18
Density [kg/m^3^]	*ρ_plate_*	2720	*ρ_p_*	7500	7500
piezoelectric strain constant [m/V]	-	-	*d_31_*	−125 × 10^−12^	−274 × 10^−12^
piezoelectric stress/charge constant [C/m^2^]	-	-	*e_31_*	-	−23.38

**Table 2 sensors-19-00812-t002:** The value of modal electromechanical coupling factor determined for the five lowest natural frequencies.

Mode Shape	The Electromechanical Coupling Factor ${\tilde{\Gamma }}_{mn}$
1st	−0.1231 × 10^−4^
2nd	−0.0331 × 10^−4^
3rd	−0.0593 × 10^−4^
4th	−0.3096 × 10^−4^
5th	−0.0604 × 10^−4^

**Table 3 sensors-19-00812-t003:** Values of the control forces generated by the piezo-actuators PA_1 and PA_2 located on the top surface of the aluminum plate.

Mode Shapes	The Amplitude of Excitation Force [N]	
	*f_n_*(*t*)_1_—for Piezo-Actuator PA_1	*f_n_*(*t*)_2_—for Piezo-Actuator PA_2
1st	404.40	−4.884
2nd	99.23	0.6469
3rd	70.522	−0.9214
4th	−33.45	4.762
5th	−113.69	−2.319

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
