# Peer review of "Analytical Modeling and Experimental Validation of an Energy Harvesting System for the Smart Plate with an Integrated Piezo-Harvester"

_sensors, 2019, doi:10.3390/s19040812_

Round 1

Reviewer 1 Report

The paper is scientifically sound but there are some overall changes that are necessary.

a) the theory of plates are well known so that vibration results are available. all the author is dong in including the harvester - the author must put the novelty better into context- why should the readers care about the work? 

b) section 4 on equivalent circuit is really weak and should be deleted.

c) Please check the English comprehensively and ensure that your presentation of references is correct. Avoid 'he' 'she' type words or 'x's group' etc. Please stick to the content and science.

d) The results for circuits change a lot with the design of the circuits. I would like to see an attempt of at least trying to optimize the circuit, even if it is simple. There are works by Erturk on this (even in the book).

e) The authors should carry out significantly more extended literature review on smart plate and present his work in the context of those published works. Significant work is necessary here.

f) Some applications or potential of that can probably make the work stronger. For example, for a plate I can think of recent works on bridge-vehicle interaction and related energy harvesting, some of which has been used for structural health monitoring and modal identification (see ASCE, MSSP). These have experimental and full-scale examples along with numerical aspects and the work can link to these or similar/other applications to give the contributions a context.

Since significant work is necessary, I recommend major revisions.

Author Response

Response to Reviewer 1 comments are in attachmnent.

Reviewer 2 Report

This paper developed an analytical electroelastic model of a piezoelectric harvester bonded on a plate with a SFSF boundary condition. The Kirchoff plate theory is mainly employed to derive the analytical model. In addition equivalent parameters of Electrical Circuit Model were found using the electroelastic model. Experimental validation was performed by comparing the predicted results with experimental results.

1.This paper has some major issues to be accepted for publication at its current format. The 2D electroelastic model using Kirchoff plate theory is already researched in the paper, (H. Yoon et al., “Kirchhoff plate theory-based electromechanically-coupled analytical model considering inertia and stiffness effects of a surface-bonded piezoelectric patch,” Smart Material and Structures, Vol.25, 2016). The technical difference between author’s work and the research in the reference should be clarified.  

2.The parameters of Electrical Circuit Model is only applicable to the boundary condition, SFSF. It is needed to get the parameters generally applicable regardless of boundary conditions, size of plates, etc.  

3.In figure 11, the values of experiments at time 0 second are not zero (0), but, the simulation results are zero(0). What is the reason of this difference?

4.In addition, the paper is not well-written. There are some awkward sentences and grammar errors in the paper. For example, meaning of the sentence, “The determined harvester location on the thin and calculated modal values of electromechanical coupling factors allow to obtain voltage ~” is not clear. It is required to carefully proofread the paper again.

5.In Table 1, “Density [kg/m3]" should be changed to “Density [kg/m3]”

Author Response

Response to Reviewer 2 comments are in attachmnent.

Round 2

Reviewer 1 Report

This version of the paper is more improved than the last version. However, the following need to be addressed:

a) The authors, for a different reviewer have considered some changes  or difference  from their paper - but has not explained why this paper is important for the readers as compared to the older paper (e.g. the authors say how the boundary conditions are different etc. but do not say wy we should care about it)

b) Reference 18 cites a paper as accepted - but it does not have a DOI and cannot be found. Consequently, it should be deleted.

c) The discrepancy for experiment is very large - especially as you go closer to zero. While the authors have given some explanation - it is probably better to include a model that captures that explanation and create a curve that is closer to the experimental observation. Otherwise the exercise is not rigorous.

Overall I am happy to consider this paper following the suggested revisions.

Author Response

Answers to yours comments in the attachment

Reviewer 2 Report

The authors responded to all my comments and updated the manuscript accordingly.

I recommend the manuscript to be accepted with a minor text editing.

(1) " 3th mode" in line 387 should be changed to "3th mode"

Author Response

Answer to your comment in the attachment
